# Development of a Polymersome-Based Nanomedicine for Chemotherapeutic and Sonodynamic Combination Therapy

**DOI:** 10.3390/ijms24021194

**Published:** 2023-01-07

**Authors:** Mingyun Kim, Doyeon Kim, Yongho Jang, Hyounkoo Han, Seonock Lee, Hyungwon Moon, Jungho Kim, Hyuncheol Kim

**Affiliations:** 1Department of Chemical and Biomolecular Engineering, Sogang University, 35 Baekbeom-ro, Mapo-gu, Seoul 04107, Republic of Korea; 2Department of Life Science, Sogang University, 35 Baekbeom-ro, Mapo-gu, Seoul 04107, Republic of Korea; 3R&D Center, IMGT Co., Ltd., Seongnam-si 13605, Republic of Korea

**Keywords:** sonodynamic therapy, combination therapy, polymersomes, drug delivery, doxorubicin, verteporfin, ultrasound

## Abstract

In anticancer therapy, combination therapy has been suggested as an alternative to the insufficient therapeutic efficacy of single therapy. Among combination therapies, combination chemo- and photodynamic therapy are actively investigated. However, photodynamic therapy shows a limitation in the penetration depth of the laser. Therefore, sonodynamic therapy (SDT), using ultrasound instead of a laser as a trigger, is an upcoming strategy for deep tumors. Additionally, free drugs are easily degraded by enzymes, have difficulty in reaching the target site, and show side effects after systemic administration; therefore, the development of drug delivery systems is desirable for sufficient drug efficacy for combination therapy. However, nanocarriers, such as microbubbles, and albumin nanoparticles, are unstable in the body and show low drug-loading efficiency. Here, we propose polylactide (PLA)-poly (ethylene glycol) (PEG) polymersomes (PLs) with a high drug loading rate of doxorubicin (DOX) and verteporfin (VP) for effective combination therapy in both in vitro and in vivo experiments. The cellular uptake efficiency and cytotoxicity test results of VP-DOX-PLs were higher than that of single therapy. Moreover, in vivo biodistribution showed the accumulation of the VP-DOX-PLs in tumor regions. Therefore, VP-DOX-PLs showed more effective anticancer efficacy than either single therapy in vivo. These results suggest that the combination therapy of SDT and chemotherapy could show novel anticancer effects using VP-DOX-PLs.

## 1. Introduction

Cancer is one of the leading causes of death worldwide. Despite efforts to overcome various types of cancer, cancer continues to increase in prevalence [1]. Common anticancer therapies currently applied to patients include chemotherapy, radiotherapy, and surgical resection. Among them, photodynamic therapy (PDT) shows high potential as a local therapy since the therapy is performed in a specific region by external triggers. PDT is an anticancer therapy that can selectively kill cancer cells with reactive oxygen species (ROS) by activating photosensitizers using light at a specific wavelength. When the photosensitizer is activated by light, the excited photosensitizer falls to the ground state and releases energy to the molecules nearby. Then, oxygen molecules that absorbed the released energy convert into ROS, which shows high cell toxicity [2]. However, PDT is limited to tumor cells near the skin due to the limitation of the depth of transmission of the light used [3]. Sonodynamic therapy (SDT), a similar therapy technique to PDT, is expected to overcome the limitation of PDT since it uses ultrasound as an energy source [4]. Though, to date, the mechanism of SDT is not clear, it is known that the sonosensitizer is activated by ultrasound and turns oxygen molecules into ROS [5]. Additionally, SDT is considered a more advanced therapy than PDT in terms of the fact that it both induces apoptosis and improves anti-tumor immunity [6]. Furthermore, combination therapy is emerging to enhance the therapeutic efficacy to overcome the insufficient anticancer effect of single therapy. Various studies have been conducted on combinations of different therapies, such as gene therapy, chemotherapy, and PDT [7,8]. Combination therapy is meaningful in that it can simultaneously exert the advantages of each treatment.

Doxorubicin, which intercalates into the DNA double-strand and inhibits the synthesis of DNA, is often selected as a chemotherapeutic drug [9]. In addition, verteporfin is used as a sonosensitizer that generates ROS in response to ultrasound. Many studies have verified the efficacy of combination therapy using drugs, such as chemotherapeutic agents and photo or sonosensitizers. However, the drugs that are used in the combination therapy are themselves vulnerable to biological conditions due to enzymes and various elements in the blood, leading to drug degradation, which lowers the anticancer effect. Therefore, several nanocarriers that can protect and deliver drugs safely by encapsulating them have been investigated [10]. For example, microbubbles [11], albumin nanoparticles [12], polymeric nanoparticles [13], and nanoemulsions [14] have been developed as nanocarriers for loading various therapeutic drugs. Among them, polymersomes are emerging as novel drug carriers. Polymersomes are liposome-like particles made by the self-assembly of amphiphilic copolymers. They are composed of hydrophobic double layers and a hydrophilic core part, enabling the simultaneous carrying of both hydrophobic and hydrophilic agents in the particles. Compared to liposomes that have a similar structure, polymersomes have strength in terms of relatively low leakage and high stability. They can also release drugs in a sustainable manner in response to external stimuli, such as pH, ultrasound, or temperature [15]. Thus, polymersomes are being actively used in many studies [16,17,18,19]. However, there are not many studies on combination therapy using polymersomes; for those that have been reported, the drug loading efficiency is low.

In this study, we developed verteporfin and doxorubicin co-loaded polymersomes (VP-DOX-PLs) composed of PLA-PEG (2k–1k) di-block copolymers to simultaneously deliver both hydrophobic and hydrophilic drugs with a single nanocarrier (Figure 1). Verteporfin, which is a hydrophobic drug, was loaded into the hydrophobic shell, and doxorubicin, which is a hydrophilic drug, was loaded into the hydrophilic core. When the polymersomes were exposed to ultrasound, ROS were generated by the sonodynamic effect of verteporfin followed by the released doxorubicin. We demonstrated through in vitro and in vivo experiments that the therapeutic efficacy of this combination therapy is superior to that of single SDT or chemotherapy, respectively. Our results indicate that the VP-DOX-PLs have high potential as novel carriers showing effective anticancer therapy through combined SDT and chemotherapy so that they can overcome the unsatisfactory anticancer effect of conventional single therapy while delivering anticancer agents safely to target sites.

## 2. Results

### 2.1. Characteristics of VP-DOX-PLs

Verteporfin and doxorubicin-loading PLA-PEG polymersomes (VP-DOX-PLs) were formed by a solvent switch method (Table 1). The average sizes of the bare PLA-PEG polymersomes (PLs), verteporfin loading PLA-PEG polymersomes (VP-PLs), and VP-DOX-PLs were ((36.43 ± 0.19), (44.72 ± 0.33), and (79.71 ± 5.62)) nm, respectively (Table 2, Figure 2A). The size of the polymersome was increased in the doxorubicin-loading process by the pH gradient method. Transmission electron microscopy (TEM) imaging shows the morphology of polymersomes composed of a hydrophobic double layer and hydrophilic core (Figure 2A, Appendix A). The encapsulation of verteporfin into polymersomes was confirmed by ultraviolet-visible spectroscopy. The absorption peak of VP-PLs was red-shifted 20 nm compared to the free verteporfin, showing successful verteporfin loading (Figure 2B). Additionally, the in vitro release profile with or without ultrasound of doxorubicin and verteporfin was confirmed by HPLC (Figure 2C,D). Both doxorubicin and verteporfin showed sustained release up to 24 h. Approximately (85 and 90)% of encapsulated doxorubicin was released at 24 h with and without ultrasound, respectively. Approximately 40% of the encapsulated verteporfin was released without or with ultrasound at 24 h. The ultrasound irradiation was found not to affect the in vitro release rate of either doxorubicin or verteporfin, demonstrating no change in the integrity of the polymersome membrane by ultrasound. The stability of the drug carrier is one of the most important elements for drug delivery. Therefore, the stability of VP-DOX-PLs was confirmed in PBS solution incubated at (4 or 37) °C (Figure 2E). After 48 h of incubation, both (4 and 37) °C incubated VP-DOX-PLs showed no significant change in mean size, demonstrating the sufficient stability of the VP-DOX-PLs. Furthermore, the VP-DOX-PLs exhibited a stable size in a DMEM + 10% FBS solution mimicking biological conditions for 48 h, indicating that in vivo injected VP-DOX-PLs could circulate for 2 days (Figure 2F).

### 2.2. Intracellular Uptake and Drug Delivery Effects of VP-DOX-PLs

To confirm whether VP-DOX-PLs could successfully deliver drugs into the cell, the intracellular uptake efficiency was investigated both qualitatively and quantitatively (Figure 3 and Appendix A). As shown in Figure 3A, either doxorubicin or verteporfin loaded in polymersomes were successfully delivered into the cell for DOX-PLs and VP-PLs groups, respectively. They entered the cell by endocytosis, and some of them escaped from the endosome, delivering drugs into the cytoplasm. When VP-DOX-PLs were treated in cells, the fluorescence of doxorubicin and verteporfin were both shown in the cells, indicating both drugs were delivered without interrupting each other’s delivery efficiency. As both doxorubicin and verteporfin have fluorescent characteristics, quantitative analysis of cellular uptake efficiency was performed by flow cytometry according to the fluorescence of doxorubicin and verteporfin (Figure 3B,C). Flow cytometry results revealed that compared to the untreated group, the mean fluorescence of both doxorubicin and verteporfin increased, indicating the co-delivery of doxorubicin and verteporfin was successful.

### 2.3. ROS Generation and Cell Viability of VP-DOX-PLs

Verteporfin, a representative sonosensitizer, is known to generate ROS in response to ultrasonic energy to kill surrounding cells [20]. To confirm whether verteporfin-loaded polymersomes could generate sufficient ROS under the exposure of ultrasound for sonodynamic therapy, a DCFDA cellular ROS detection assay was conducted (Figure 4A). Before the irradiation of ultrasound, there was no statistically significant difference in ROS levels between all groups. However, when irradiated with ultrasound, the VP-DOX-PLs group showed approximately 1.73 times higher fluorescence intensity compared to the control, while other groups showed no increase in fluorescence. As VP-DOX-PLs successfully delivered verteporfin to cells, ROS was generated through the sonodynamic effect of verteporfin. This result suggests that effective sonodynamic therapy (SDT) can be performed with VP-DOX-PLs.

Furthermore, in vitro cytotoxicity experiments were conducted to determine the efficacy of combination chemotherapy and sonodynamic therapy (Figure 4B). The PLs-treated group (polymersomes without drugs) did not show significant cytotoxicity, demonstrating no toxicity of PLs itself and no effect of ultrasound on cell viability. The DOX-PLs treated group showed approximately 40% cell viability compared to the control group, regardless of ultrasound exposure, which is caused by the chemotherapeutic effect of doxorubicin. In the case of the VP-PLs treated group, there was no significant cell death before ultrasound irradiation. However, after exposure to ultrasound, more than 30% of cells were dead because of ROS generation from verteporfin. Most importantly, the VP-DOX-PLs-treated group, the combination therapy group, showed similar results to that of the DOX-PLs without ultrasound but the highest cytotoxicity after ultrasound irradiation. In addition, (3 and 6) h uptake experiments were conducted to investigate the cytotoxicity efficiency according to the uptake time of the particles (Appendix A). The cell viability was approximately 80% at 3 h, 60% at 6 h, and decreased to 20% at 24 h. As the uptake time increased from (3 to 24) h, the efficacy of combination therapy increased. Cell cytotoxicity data reveal that the therapeutic efficacy of the combination therapy of chemotherapy and SDT was superior to all the other single therapies and increased according to the uptake time.

### 2.4. In Vivo Biodistribution of VP-DOX-PLs

To verify the biodistribution of intravenous injected VP-DOX-PLs, non-invasive whole-body imaging was used to assess the in vivo biodistribution. The fluorescence of verteporfin was imaged with FOBI 24 h post-injection. The VP-DOX-PLs-injected group showed an accumulation of PLs in the tumor site by the EPR effect, indicating that the VP-DOX-PLs were successfully delivered into the tumor region (Figure 5A). After 24 h post-injection, major organs and tumor were excised to compare the distribution of the PLs in the tumor-bearing mice (Figure 5B,C). The verteporfin fluorescence signals of the tumor organ were the highest, whereas the verteporfin fluorescence signals of the other organs showed low fluorescence signals. This result indicates that the size of VP-DOX-PLs was effective for accumulation at the tumor site in in/ex vivo.

### 2.5. Anti-Tumor Effects of VP-DOX-PLs with Ultrasound Irradiation

To determine the superior therapeutic efficacy of VP-DOX-PLs with ultrasound irradiation in comparison to single therapy (DOX-PLs or VP-PLs), DOX-PLs, VP-PLs, or VP-DOX-PLs were intravenously administered to MIA PaCa-2 tumor-bearing mice. The tumor volume was measured to determine the therapeutic efficacy (Figure 6A). First, VP-PLs (−) did not show statistically significant therapeutic efficacy compared to the control group, as insufficient ROS were generated without an ultrasound trigger. DOX-PLs and VP-DOX-PLs (−) showed similar therapeutic efficacy, showing reduced tumor growth by chemotherapy. VP-PLs (+) also showed similar therapeutic efficacy to DOX-PLs, indicating single chemotherapy or SDT has a similar therapeutic effect. VP-DOX-PLs (+) showed the highest therapeutic efficacy on tumor growth inhibition. Additionally, the extracted tumor tissues from each group were imaged, and the weight was measured (Figure 6B,C). Among all groups, VP-DOX-PLs (+) showed the lowest weight consistent with the smallest tumor volume. Furthermore, histological assays, including hematoxylin and eosin (H&E) and the TUNEL assay, were performed (Figure 6D). Clear cell nuclei and normal morphology were observed in the negative control group. Damaged cell nuclei were observed in either the chemotherapy or sonodynamic therapy group. VP-DOX-PLs groups with ultrasound exposure showed much more damaged cell nuclei and pore in the tumor, indicating higher therapeutic efficacy. To monitor the side effects of the systemic administration of VP-DOX-PLs, the body weight of mice was measured (Figure 6E). No group showed a significant change in body weight, revealing no severe cytotoxicity of VP-DOX-PLs. Additionally, H&E staining on the major organs revealed that no treated group showed any significant toxicity on major organs (Appendix A). In conclusion, the combination therapy of chemo and sonodynamic therapy demonstrated the strongest therapeutic efficacy compared to the single therapy groups.

## 3. Discussion

In this study, we developed VP-DOX-PLs for the combination of chemotherapy and sonodynamic therapy, resulting in increased therapeutic efficacy compared to single therapies. Two drugs, doxorubicin and verteporfin, are loaded into polymersomes. Doxorubicin, which is commonly used as an anticancer drug, enters the nucleus of the cell and binds to DNA, arresting cell division. Verteporfin is one of the sonosensitizers for sonodynamic therapy. A sonosensitizer absorbs ultrasonic energy and releases energy to the molecules around it. In this process, nearby oxygens excite and become radicals, known collectively as ROS, which damage DNA or proteins in the cell, leading to the apoptosis of the cell. The developed VP-DOX-PLs drug delivery system could deliver drugs into tumors through the following steps: (1) development of doxorubicin and verteporfin co-delivering polymersomes; (2) enhanced tumor accumulation by EPR effect and the intracellular delivery of doxorubicin and verteporfin using polymersomes; (3) the local treatment of sonodynamic therapy due to ultrasound irradiation.

In the first step, chemotherapeutic agents (doxorubicin) and a sonosensitizer (verteporfin) were loaded in different parts of the polymersome. Polymersomes are self-assembled nanoparticles that are known to have superior stability to that of other particles and show relatively low leakage of loaded drugs [18]. Polymersomes are capable of loading hydrophilic drugs within the aqueous core of the polymersomes and hydrophobic drugs loaded within the membrane bilayer. Therefore, polymersomes possess broad possibilities in drug delivery [21,22] and gene and protein delivery [19] systems for therapeutic application. Since polymersomes are developed in a thermodynamically stable way, various particles, such as micelles and polymeric particles, can be formed, as well as polymersomes. The packing parameter, $P=v\left(a\times l\right)$, is commonly used to predict the morphology of the self-assembled formulations (where l is the length of the block copolymer, a is the area of the hydrophilic part, and v is the volume of the hydrophobic part). Other than this method, the morphology can be roughly predicted through the molecular weight ratio of the hydrophilic polymer ($f=M_{hydrophilic}/M_{total\;polymer})$ [18]. Polymersomes are known to be formed when 0.25<f<0.4. From this point of view, PLA (2k)-PEG (1k) copolymer has a molecular weight that is sufficient for the self-assembly of polymersomes in solution. In general, the bilayer of polymersomes could be clearly observed by TEM imaging; however, due to the low molecular weight of the PLA-PEG (2k–1k) used, the boundary of the double membrane was ambiguous (Figure 2A). To investigate whether the core part of VP-DOX-PLs is hydrophilic, hydrophilic 5 nm gold nanoparticles were loaded into polymersomes. TEM imaging of gold nanoparticle-loaded VP-DOX-PLs revealed that gold nanoparticles were located inside the core part of VP-DOX-PLs, indicating that the core part is hydrophilic (Appendix A). Therefore, in the case of VP-DOX-PLs, hydrophilic doxorubicin was in the core of the polymersomes, and hydrophobic verteporfin was located in the double membrane of the polymersomes. Doxorubicin was loaded into PLs using the pH gradient method, which is an active loading method of drugs according to the transmembrane pH outside and inside the liposome [23]. Since doxorubicin enters the polymersome core and forms (DOX-NH_3_)_2_SO_4_ crystallization, it showed 95.05% loading efficiency. Compared to that of passive loading, the pH gradient method showed approximately 1.8 times higher loading efficiency. The stability of VP-DOX-PLs was evaluated by measuring the size for 48 h (Figure 2E,F). As the initial burst of both drugs is carried out during the first 24 h (Figure 2C,D), the stability of VP-DOX-PL is sufficient to deliver the drug to the tumor.

In the second step, once VP-DOX-PLs enter the body through intravenous injection, they accumulate near tumor sites by the EPR effect [24]. The dense PEG corona of the PEG copolymer vesicles was shown to deter membrane opsonization and significantly extend in vivo circulation times [25]. Due to the extended circulation time by the PEG surface, VP-DOX-PLs were successfully delivered into the tumor region by passive targeting. Nanoparticles larger than 10 nm generally cannot penetrate the blood vessels and enter the cells from normal blood vessels. However, abnormal blood vessels are formed around cancer cells because cancer cells secrete a lot of blood vessel growth factors. These blood vessels have looser walls than normal blood vessels, allowing larger nanoparticles to penetrate the vessels and enter the cells. The in vivo biodistribution of VP-DOX-PLs reveals that VP-DOX-PLs were able to accumulate at the tumor site at 24 h post-injection (Figure 5). The fluorescence of verteporfin of tumor tissue was highest compared to the other major organs, especially the liver and kidney. In this process, another function of ultrasound could be affected. When ultrasound is irradiated, the blood vessel oscillates, and the permeability of the vessel increases, which is called sonoporation. Next, VP-DOX-PLs located near tumor tissue need to enter the cell. The in vitro cellular uptake results showed that VP-DOX-PLs successfully delivered both verteporfin and doxorubicin into cells. As seen in Figure 3, the fluorescence of doxorubicin was spread throughout the cell, while verteporfin’s fluorescence was found near the nucleus of the cell. To check whether VP-DOX-PLs directly penetrate the nucleus through the new mechanism, rather than by endocytosis or the fusion of nanoparticles, the cellular uptake of the particles was analyzed in 3D using confocal microscopy (Appendix A). The fluorescence of verteporfin did not penetrate the nucleus but was located near the nucleus. It is estimated that the particles entered the Golgi apparatus through clathrin-dependent endocytosis [26]. Since doxorubicin is hydrophilic, it spread throughout the cell, but hydrophobic verteporfin stayed in the Golgi apparatus. However, there is no clear explanation of the exact cell uptake mechanism of VP-DOX-PLs, which needs to be further investigated.

In the third step, after the accumulation of VP-DOX-PLs on the tumor site by passive targeting, the ultrasound irradiation specific to the tumor site enables the local treatment of sonodynamic therapy. The results of combination therapy showed more effective cancer treatment than other single therapies in both the in vitro and in vivo experiments. To confirm the cytotoxic effect according to the uptake time of the particles, cytotoxicity experiments were performed at various time points of (3, 6, and 24) h (Figure 4B, and Appendix A). The cell viability of the PLs group showed no significant difference from that of the control, meaning that the PL itself is non-toxic. VP-DOX-PLs without ultrasound and DOX-PLs showed similar anticancer effects by arresting DNA replication and the cell division of doxorubicin. Moreover, as the incubation time of particles increased, cell viability was decreased in groups loading doxorubicin. This result indicates chemotherapy does not show an instant response in viability, while sonodynamic therapy is instant. Consistent with the in vitro results, the therapeutic efficacy of both drugs using polymersomes with ultrasound was also determined by the in vivo mouse model. After the accumulation of VP-DOX-PLs, cell apoptosis of cancer cells occurs by DNA repair disruption due to the intercalation of doxorubicin. Furthermore, when ultrasound is irradiated to the local tumor region, the sonodynamic therapy effect occurs, and ROS is generated by the verteporfin [27,28]. Note that the VP-PLs and VP-DOX-PLs exposed to ultrasound showed higher cytotoxicity than those without ultrasound (Figure 4B), indicating ultrasound is a trigger for sonodynamic therapy. Therefore, off-targeted VP-DOX-PLs did not damage other organs (Appendix A). In addition, verteporfin was used as a sonosensitizer, but it is also a photosensitizer. The difference between sonodynamic therapy and photodynamic therapy is the trigger of therapy: ultrasound for sonodynamic therapy and laser for photodynamic therapy. However, as it is easier for an ultrasound to non-invasively reach deep tissues and activate an accumulated sonosensitizer than a laser, ultrasound is advantageous for treating cancers that occur deep inside the body, such as pancreatic cancer [29].

## 4. Materials and Methods

### 4.1. Materials

Poly-L-lactide-poly (ethylene glycol) was purchased from Creative PEG Works (Chapel Hill, NC, USA). Verteporfin, tetrahydrofuran, acetonitrile, and other reagents were purchased from Sigma–Aldrich (St. Louis, MO, USA). Doxorubicin was purchased from MedChemExpress (Monmouth Junction, NJ, USA). DMEM, fetal bovine serum (FBS), and antibiotic–antimycotic (AA) were purchased from Welgene (Gyeongsan, Korea). MIA PaCa-2 cancer cells were sourced from the Korean Cell Line Bank (Seoul, Korea). All other chemicals and solvents were analytical grade.

### 4.2. Synthesis of VP-DOX-PLs

Both verteporfin and doxorubicin-loading PLA-PEG polymersomes (VP-DOX-PLs) were formed by a solvent switch method. Briefly, 20 mg of PLA-PEG copolymers (PLA (2k)—PEG (1k)) and 0.02 mg of verteporfin were dissolved in 1 mL of tetrahydrofuran (THF). Then, the polymer solution was slowly dropped into 1 mL of water at a ratio of 1 mL/h to form only verteporfin-loading PLA-PEG polymersomes (VP-PLs). The polymer solution was slowly stirred at 25 °C overnight to remove the organic solvent THF. After, the free verteporfin was removed by centrifugation using a 3 kDa amicon tube (Millipore, Burlington, MA, USA) for 30 min at 14,000 g and redispersed in distilled water. Doxorubicin was loaded into VP-PLs by the ion gradient method. Doxorubicin hydrochloride (1 mg/mL) was added to 1 mL of 250 mM ammonium sulfate solution. Then, VP-PLs were reacted with doxorubicin solution at 400 rpm for 2 h at 25 °C. Unloaded doxorubicin was removed using 3 kDa amicon tubes again. Finally, VP-DOX-PLs were redispersed in distilled water.

### 4.3. Characterization of VP-DOX-PLs

The size distribution of the particles was analyzed using a dynamic light scattering device (Zetasizer Nano, Malvern, Malvern Instruments, Herrenberg, Germany). The morphological characteristics of VP-DOX-PLs were determined by transmission electron microscopy (TEM; JEM-2100F, JEOL, Tokyo, Japan). The encapsulated efficiency of doxorubicin was measured using reversed-phase high-performance liquid chromatography (Agilent HPLC 1200, Agilent, Santa Clara, CA, USA). Additionally, the encapsulation efficiency of verteporfin was measured using UV–Vis spectrophotometry (Evolution^™^ 60S, Thermo Scientific, East Grinstead, UK). The in vitro release rate of either doxorubicin or verteporfin from VP-DOX-PLs was measured by placing VP-DOX-PLs in a dialysis membrane (Spectra/Por^®^ 7, MWCO 2 kD, SPECTRUMLABS, USA), and incubation with 10 mL phosphate-buffered saline (PBS) at 37 °C in shaking conditions. At the predetermined time points, incubated 10 mL of the PBS buffer was replaced with fresh PBS buffer, and the amount of the released drug was measured through HPLC or UV–Vis spectrophotometry.

### 4.4. In Vitro Stability of VP-DOX-PLs

The in vitro stability of VP-DOX-PLs depending on temperature was analyzed by the change in the size of VP-DOX-PLs through a dynamic light scattering device (Zetasizer Nano, Malvern, Malvern Instruments, Herrenberg, Germany). VP-DOX-PLs were incubated at (4 or 25) °C in PBS solution. Additionally, the stability of VP-DOX-PLs was determined in DW and 10% FBS solution (in DMEM) at 37 °C. After a predetermined time, the size of VP-DOX-PLs was measured through DLS until 48 h for both experiments.

### 4.5. Intracellular Uptake of VP-DOX-PLs

Human pancreatic cancer cells (MIA PaCa-2 cells, Korean Cell Line Bank, Seoul, Korea) were maintained in RPMI-1640 containing 10% fetal bovine serum and 1% penicillin/streptomycin at 37 °C in a humidified atmosphere of 5% CO_2_. To visualize the intracellular delivery efficiency of VP-DOX-PLs, MIA PaCa-2 cells (2 × 10^5^ cells per well) were seeded in confocal dishes. Doxorubicin and verteporfin were treated at the concentration of (200 and 100) ng/mL, respectively, and incubated for 24 h. The cells were washed twice using cold PBS and fixed using 4% paraformaldehyde for 15 min. Then, the cells were stained with DAPI solutions, and fluorescence images of DOX-PLs, VP-PLs, and VP-DOX-PLs were obtained using confocal laser scanning microscopy (LSM 880, Zeiss, Jena, Germany). For the quantitative analysis, the same experiment was conducted with Accuri C6 Plus (BD Accuri C6 Plus, BD, Piscataway, NJ, USA), and 10,000 cells were counted for each group. The fluorescence of doxorubicin was measured through the FITC filter, and the fluorescence of verteporfin was measured using an APC filter.

### 4.6. Confirmation of ROS Generation under the Exposure of Ultrasound

The generation of reactive oxygen species (ROS) was confirmed under the exposure of ultrasound to determine the sonodynamic therapy potential of VP-DOX-PLs. MIA PaCa-2 cells were seeded in 96-well plates (1 × 10^4^ cells per well) and incubated for a day. Empty PLs (no verteporfin and no doxorubicin), DOX-PLs, and VP-DOX-PLs were treated in the cells. Each group was treated with the same amount of verteporfin (a concentration of 1 μg/mL of verteporfin). Ultrasound was irradiated with 0.3 W/cm^2^ ultrasonic power and a 50% duty cycle for 10 s per well using Sonopuls 490 (Enraf Sonopuls, Harlev, Denmark). The amount of reactive oxygen species (ROS) generation of MIA PaCa-2 cells was measured using a DCFDA Cellular ROS Detection Assay Kit (Abcam, Cambridge, UK), according to the manufacturer’s protocol. The fluorescence intensity was measured using a microplate reader (EnSpire multimode plate reader, PerkinElmer, Waltham, MA, USA) at Ex/Em = 485/535 nm/nm.

### 4.7. In Vitro Cell Viability following Ultrasound Irradiation

Human pancreatic cancer cells (MIA PaCa-2 cells) were sourced from the Korean Cell Line Bank (Seoul, Korea). To verify the therapeutic effect of chemotherapy and sonodynamic therapy, an in vitro cell viability test was performed using the MTT assay. MIA PaCa-2 (2 × 10^4^ cells per well) was seeded in 96-well cell culture plates and incubated for 24 h (37 °C, 5% CO_2_). Ultrasound was irradiated to each group at 0.3 W/cm^2^, 50% duty cycle, 10 s per well, and further incubated for (3, 6, and 24) h. The concentration of doxorubicin and verteporfin was (200 and 100) ng/mL, respectively. After the incubation of each treated group, the cells were washed twice using cold PBS, exchanged into fresh medium, and further incubated for 24 h. Next, 30ul of MTT solution (0.5 mg/mL) were added to each well, and after 3 h incubation, DMSO (200 μL) was added. The absorbance was measured at a 595 nm wavelength by a microplate reader (Bio-Tek, Winooski, VT, USA).

### 4.8. In Vivo Studies

All in vivo studies conformed to the Guide for the Care and Use of Laboratory Animals published by the National Institutes of Health, USA (NIH publication no. 85-23, 1985, revised 1996). Mice were maintained under the guidelines of an approved protocol from the Institutional Animal Care and Use Committee (IACUC) of Sogang University (Republic of Korea, Approval Date: 01/20/2021 Approval Code: IACUCSGU2021_02).

### 4.9. In Vivo and Ex Vivo Imaging of VP-DOX-PLs in MIA PaCa-2 Tumor-Bearing Mice

To determine the accumulation of VP-DOX-PLs in the tumor, in vivo and ex vivo imaging of VP-DOX-PLs after intravenous injection was performed using the FOBI system. For the subcutaneous tumor model, 5-week-old female Balb/C nude mice were injected in the lower flank with MIA PaCa-2 cells (1 × 10^6^ cells per mouse) (Raonbio, Seoul, Korea). When the volume of tumor reached 150 mm^3^, VP-DOX-PLs were intravenously injected for a biodistribution study. The biodistribution of VP-DOX-PLs was monitored using the FOBI system through an NIR filter (Fluorescence labeled Organism Bioimaging, NeoScience, Suwon, Korea), and the intensity of the fluorescence signal was quantified by NEOimage software (NeoScience, Suwon, Korea). At 24 h post-injection, tumors and major organs, including the heart, lung, spleen, liver, and kidneys, were excised and imaged with an FOBI system.

### 4.10. Anti-Tumor Effects of VP-DOX-PLs with US Irradiation

To investigate the anti-tumor effects of VP-DOX-PLs, MIA PaCa-2 cells (1 × 10^6^) were injected subcutaneously into 5-week-old female Balb/C nude mice (n = 3, Raonbio, Seoul, Korea). When the tumor volume reached (30–50) mm^3^, each group of mice were intravenously injected at an equivalent dose of 5 mg/kg of doxorubicin and 1 mg/kg of verteporfin three times every two days. For ultrasound-positive groups, tumors were irradiated by the ultrasound (2.0 W/cm^2^, 50% duty cycle, 1 min per mouse) soon after i.v. injection. The volume of the tumor was measured by an external caliper based on the following formula: tumor volume = (length × width^2^)/2. The body weights were monitored during the experiment. At the end of the experiment, mice were sacrificed, and the tumors and major organs were examined for histological analysis. For histological analysis, a part of the excised tumors and major organs were fixed in 4% paraformaldehyde, embedded in paraffin, sectioned (5 μm), and stained with an H&E and TUNEL assay kit following the manufacturer’s protocols.

### 4.11. Statistical Analysis

All experimental data obtained from the cultured cells and animals are expressed as the mean ± standard error from at least 3 independent experiments. The statistical significance of the difference between experimental and control groups was determined by Student’s t-test. Statistical significance was established at *p* < 0.05, and significant differences are shown by asterisks in the figures.

## 5. Conclusions

Herein, we developed ultrasound responsive polymersomes that were dual-loaded with chemotherapeutic agents (Doxorubicin) and sonosensitizer (verteporfin) (VP-DOX-PLs) for combination anti-tumor therapy. The developed VP-DOX-PLs are composed entirely of biocompatible materials, such as PLA and PEG, and FDA-approved therapeutic agents, such as doxorubicin and verteporfin. The in vitro results revealed that both drugs were successfully delivered into the cells and showed sufficient cell cytotoxicity with ultrasound irradiation. The combinational therapeutic efficacy of VP-DOX-PLs was confirmed in both in vitro and in vivo experiments. Consequently, PLA-PEG polymersomes are a promising candidate as a platform for the enhanced delivery of multiple drugs and can be regarded as a novel drug delivery system for other combination therapy.

## Figures and Tables

**Figure 1 ijms-24-01194-f001:**
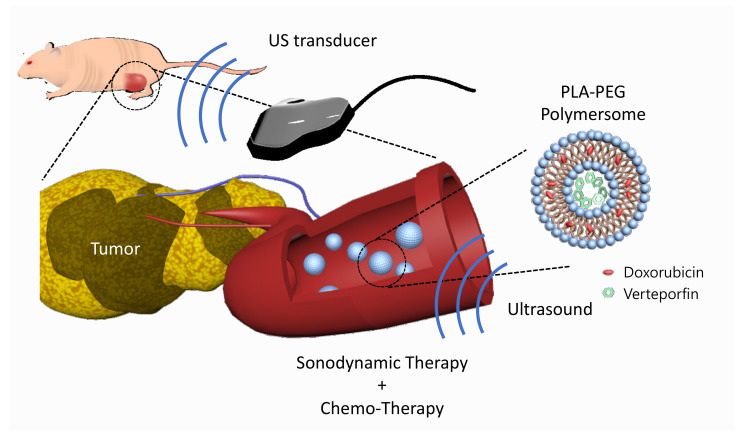
Scheme of VP-DOX-PLs. Development of the doxorubicin and verteporfin loaded polymersome for chemo and sonodynamic therapy. Blue nanoparticle represents PLA-PEG polymersome, red represents doxorubicin, and green represents verteporfin.

**Figure 2 ijms-24-01194-f002:**
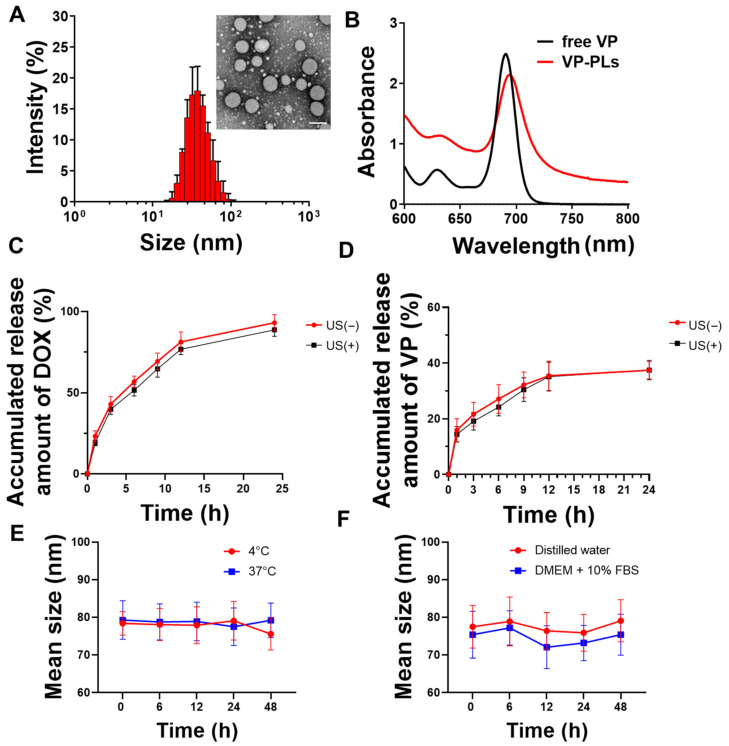
Characteristics of VP-DOX-PLs. (**A**) Size distribution and inset TEM image of VP-DOX-PLs. Scale bar indicates 100nm, (**B**) UV absorbance spectrum of free verteporfin (VP), and verteporfin loaded polymersomes (VP-PLs). (**C**) Doxorubicin and (**D**) verteporfin in vitro release profile from VP-DOX-PLs with and without ultrasound (US) irradiation. (**E**) In vitro stability of VP-DOX-PLs at (4 and 37) °C in PBS and in (**F**) DW and DMEM + 10% FBS serum solution.

**Figure 3 ijms-24-01194-f003:**
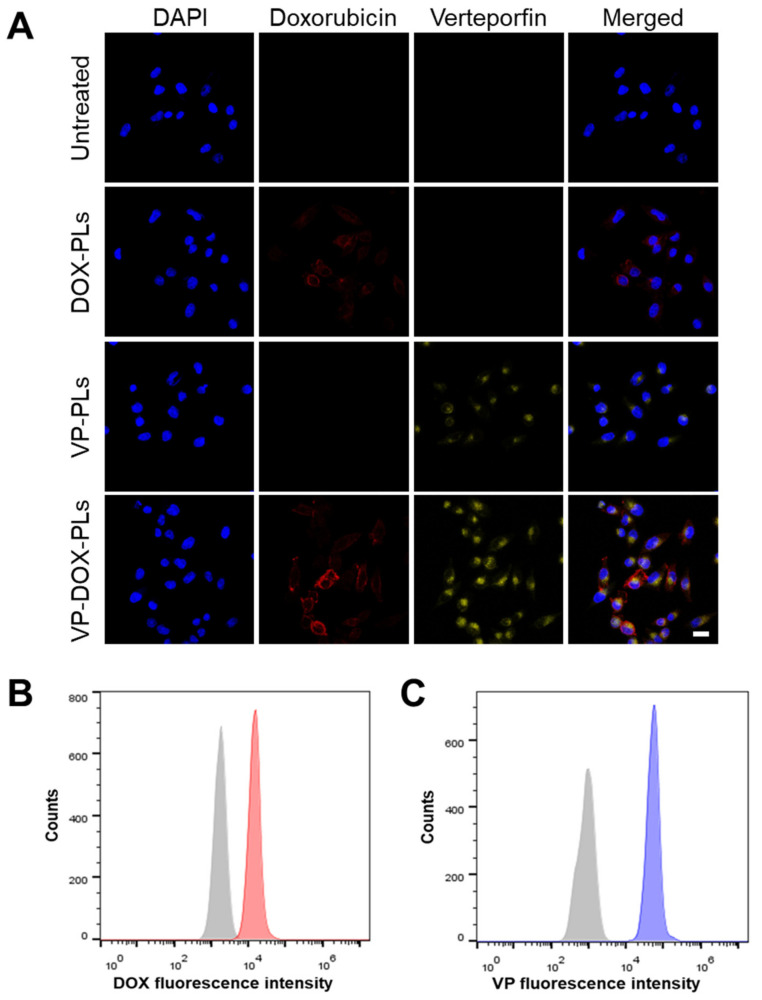
Intracellular uptake of VP-DOX-PLs. Untreated group refers to the negative control. (**A**) Confocal images of intracellular uptake of VP-DOX-PLs. Blue, red, and yellow fluorescence represent the nucleus of the cell, doxorubicin and verteporfin, respectively. Scale bar indicates 20 μm. Flow cytometry results of VP-DOX-PLs showing the fluorescence intensity of (**B**) doxorubicin (Grey represents negative control and red represents VP-DOX-PLs group) and (**C**) verteporfin (Grey represents negative control and blue represents VP-DOX-PLs group).

**Figure 4 ijms-24-01194-f004:**
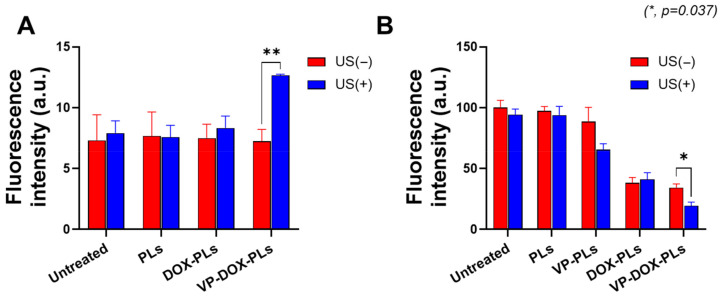
ROS generation and cell viability of VP-DOX-PLs-treated MIA PaCa-2 cells. Untreated group refers to the negative control. (**A**) ROS generation inside MIA PaCa-2 cells treated with various particles. (**B**) Cytotoxicity of VP-DOX-PLs compared with other groups with and without ultrasound. The significance of the difference between experimental and control groups was determined by Student’s t-test. * *p* < 0.05, ** *p* < 0.01.

**Figure 5 ijms-24-01194-f005:**
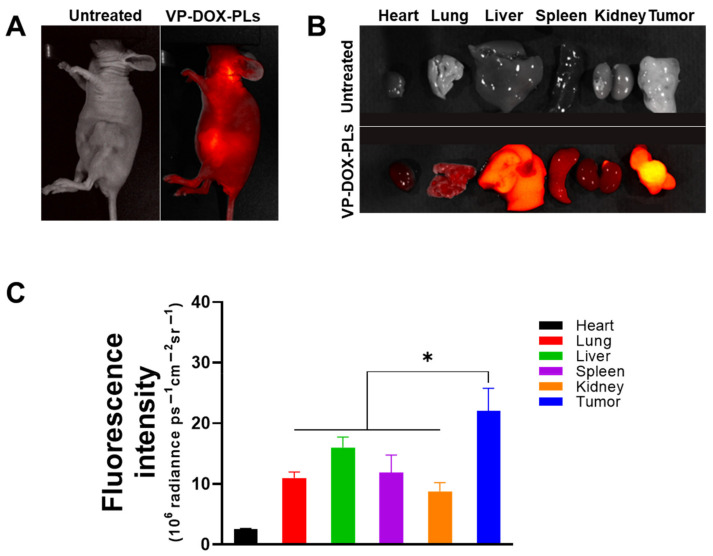
In vivo biodistribution of VP-DOX-PLs in a xenograft tumor model. (**A**) FOBI data of untreated and VP-DOX-PLs injected intravenously into the mice (Magnification 1.4x). (**B**) Fluorescence images of the major organs from the injected mice (Magnification 1.4x). (**C**) Fluorescence intensities of the major organs and tumor. The significance of the difference between experimental and control groups was determined by Student’s t-test. * *p* < 0.05.

**Figure 6 ijms-24-01194-f006:**
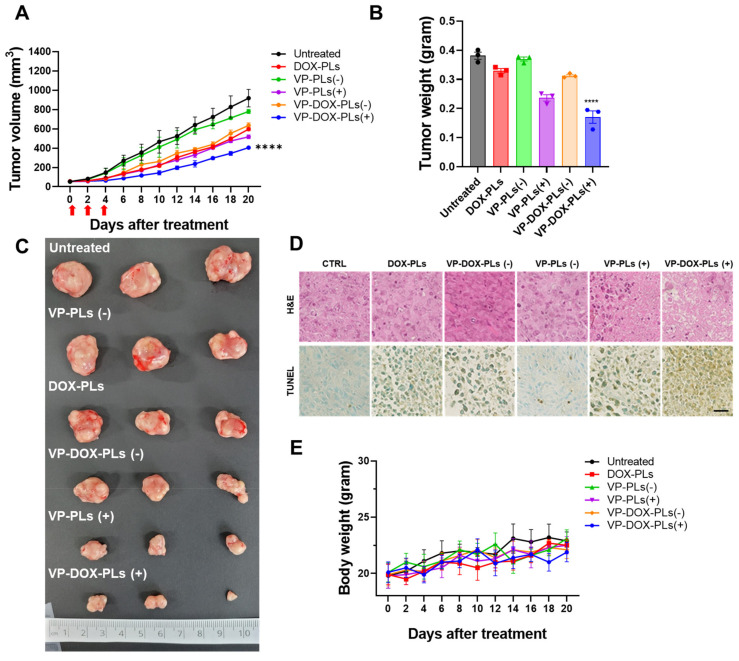
In vivo therapy of VP-DOX-PLs in a subcutaneous xenograft tumor model. (**A**) Tumor growth curve of various treatment groups. **** *p* < 0.0001. (**B**) Ex vivo tumor weights **** *p* < 0.0001 and (**C**) photographic images of different treatments on tumors after 20 days. (**D**) H&E and TUNEL assay results of tumor tissue treated in treatment groups. Scala bar indicates 20 μm. (**E**) Relative body weight change in treatment groups.

**Table 1 ijms-24-01194-t001:** Encapsulation efficiency of doxorubicin and verteporfin in DOX-VP-PLs.

	Loading Amount (mg)	E.E. (%)
Doxorubicin	1	95.05 ± 0.84
Verteporfin	0.02	100

**Table 2 ijms-24-01194-t002:** DLS data of PLs, VP-PLs, and DOX-VP-PLs.

Samples	Z-Average (nm)	PDI
PLs	36.43 ± 0.19	0.18
VP-PLs	44.72 ± 0.33	0.26
VP-DOX-PLs	79.71 ± 5.62	0.15

## Data Availability

Data are available upon reasonable request.

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
