# Peer review of "Development of a Polymersome-Based Nanomedicine for Chemotherapeutic and Sonodynamic Combination Therapy"

_ijms, 2023, doi:10.3390/ijms24021194_

Round 1
Reviewer 1 Report
The paper shows the potential of the combination therapy in the cancer treatment. The combination of two different drugs (presenting different functionalities) and ultrasound is an innovation.
Unfortunately, the paper is not well structured and it is difficult to follow and understand the results. In order to improve it so that the significance of the obtained results reaches more readers I would suggest the following changes:
1) The introductory part should deal with the state of the problem and should clearly state the research goals only. The results should be listed in the conclusions;
2) Then “Materials and methods” part (Point 4 at present) has to follow. Here step by step all materials, synthesis methods, experimental techniques and statistical estimations should be given and explained. This will fulfil two goals: will help understanding the results and will avoid the need of repeating some texts several times;
3) Only so the part “Results” will be understandable and clear;
4) Then in “Discussion” one should only discuss the main findings. At present, this part is a mixture of synthesis, experimental details and procedures and it is not clear what the authors what to underline or make clear statements;
5) The paper has to end with “Conclusions”, as it is well done at present.
Author Response
Thank you for reviewer’s comment. We agree with the review’s comment and reorganized the manuscript.
Technical comments:
The authors use a word “imagery” instead of “imaging”.
Author Response
Thank you for your comment. We replaced the term “imagery” to “imaging” for clear understanding.
In Fig,2 wavelength measuring unit is missing.
Author Response
We added the measuring units of wavelength (nm) in Fig.2 B.
In Table2 the size is given without measuring units.
Author Response
Following the reviewer’s comment, we added measuring units of z-average (nm) in Table 2.
In Figs 3 and 4 a sample “untreated” is given without a description what does it mean.
Author Response
The “untreated” group refers to negative control. We added description about untreated group in legend of Fig.2 and Fig.3.
In Fig. 6B at which day was the measurement performed?
Author Response
At the end of measurement, after 20 days, the tumor of each group was excised and weights were examined.
The References are written in different style. Should it be like that?
Author Response
Thank you for reviewer’s comment. We changed style of references according to IJMS format.
Scientific issues:
1) The US parameters may affect the results significantly since cavitation (stable or unstable), microstreaming and other phenomena appear. How do the authors choose the US parameters is in the present study?
Author Response
Thank you for reviewer’s comment. We confirmed that VP-DOX-PLs does not exhibit cavitation effect (stable or unstable). Our in vitro release drug profile without or with ultrasound reveals that polymersome are not affected by ultrasound. Therefore, we chose US parameters suitable for sonodynamic therapy for in vitro and in vivo studies.
2) In Fig.5 it is clearly seen that the fluorescence in the liver is also significant. What criterion chosen so that the authors stated the drug accumulation is significant in the tumour only?
Author Response
We agree with the reviewer’s comment. However, the main purpose of this experiment is to demonstrate the ultrasound responsive therapeutic efficacy of VP-DOX-PLs. Therefore, we stated that the drug accumulation in tumor is significant because ultrasound would be irradiated only in the tumor site.
3) How many mice were included in the experiments? The authors speak about statistics but they do not describe how big the groups were.
Author Response
Total of 18 mice were included in the experiments (n=3 for each group).
4) The authors delivered the particles with drugs when the tumours reach specific volume. How was the volume estimated?
Author Response
Tumor volume was measured by external caliper based on the following formula: tumor volume = (length × width2)/2. We added following information in section 4.10.
Reviewer 2 Report
An interesting knowledge has been proposed however the following comments should be addressed before acceptance
Minor Comments
-suggested to add some qualitative results in abstract section
Author Response
Thank you for reviewer’s comment. We included results of antitumor efficacy as well as cellular uptake and cytotoxicity of VP-DOX-PLs in abstract section.
- suggested to discuss the importance of polymeric NPs for anticancer therapy with other NPs
Author Response
Following the reviewer’s comment, we discussed the advantages of polymersome in the discussion (page 10).
-authors should discuss about the stability of both polymersomes
Author Response
We appreciate reviewer’s comment. Discussion about the stability of VP-DOX-PLs was added in page 10. We examined the stability of VP-DOX-PLs by measuring the size according to the time. No significant size change was observed for 48 h, so we concluded that the stability of VP-DOX-PLs is sufficient to deliver both drugs for antitumor therapy.
- how does the size of VP−DOX−PLs effects the antitumor effect
Author Response
The passive targeting of nanoparticles mainly relies on the EPR effect. However, bigger nanoparticles are easily eliminated by reticuloendothelial system (RES). Therefore, it is estimated that the small size of VP-DOX-PLs affected the high accumulation in tumor region rather than liver of kidney. In addition, we are also planning to discover the effect of size of polymersomes in antitumor effect.
-update reference, which are not properly formatted for this journal
Author Response
Thank you for reviewer’s comment. We changed style of references according to IJMS format.
After addressing all the comments this manuscript can be acceptable for further progress
Reviewer 3 Report
In their paper titled "Development of a polymersome-based nanomedicine for chemotherapeutic and sonodynamic combination therapy," the authors report the development of VP-DOX -PLs for the combination of chemotherapy and sonodynamic therapy and the successful evaluation of this method in vitro and in vivo. The manuscript is well written and covers an interesting topic with great relevance to cancer treatment. Some comments should be made before acceptance.
Sonodynamic therapy is not mentioned in the abstract. What is SDT in the last sentence?
Author Response
Thank you for reviewer’s comment. We added description and mentioned SDT as sonodynamic therapy in the abstract.
SDT as sonodynamic therapy is not introduced until page 7.
Author Response
We appreciate reviewer’s comment. We introduced SDT as sonodynamic therapy when first mentioned.
US in Figure 2 should be introduced in the legend.
Author Response
We appreciate reviewer’s comment. We introduced US as ultrasound when first mentioned in the Fig.2 legend.
The blue color in the images of Figure 3A should be explained in the legend.
Author Response
The blue color in the Fig.3A refers to nucleus of cell. We added following information in the legend.
How was the statistic (significant difference) determined in Figure 4 and 5? It would be better to state this in the legend.
Author Response
The significance of difference between experimental and control groups was determined by Student’s t-test. (*P<0.05, **P<0.01, ***P<0.001) Following the reviewer’s comment, we added statement about significant difference in legend of Fig.4 and Fig.5.
In Figure 6, the scale bars are missing in the H&E images and in the tunnel assay. These images are too small to see that "damaged nuclei were observed in both the chemotherapy and sonodynamic therapy groups," as the authors write. Zoomed representative images are better.
Author Response
Following the reviewer’s comment, we zoomed images of H&E and TUNNEL assay in Fig. 6D for better representative images. Also, we added scale bars and legend of Fig. 6D.